# Emergency Medical Services Calls Analysis for Trend Prediction during Epidemic Outbreaks: Interrupted Time Series Analysis on 2020–2021 COVID-19 Epidemic in Lazio, Italy

**DOI:** 10.3390/ijerph19105951

**Published:** 2022-05-13

**Authors:** Antonio Vinci, Amina Pasquarella, Maria Paola Corradi, Pelagia Chatzichristou, Gianluca D’Agostino, Stefania Iannazzo, Nicoletta Trani, Maria Annunziata Parafati, Leonardo Palombi, Domenico Antonio Ientile

**Affiliations:** 1Local Health Authority “Roma 1”, 00193 Rome, Italy; 2Azienda Regionale Emergenza Sanitaria ARES 118, 00149 Rome, Italy; apasquarella@ares118.it (A.P.); mpcorradi@ares118.it (M.P.C.); pchatzichristou@ares118.it (P.C.); gdagostino@ares118.it (G.D.); siannazzo@ares118.it (S.I.); ntrani@ares118.it (N.T.); mparafati@ares118.it (M.A.P.); daientile@ares118.it (D.A.I.); 3Department of Biomedicine and Prevention, Post-Graduate School of Specialization in Hygiene and Preventive Medicine, University of Rome “Tor Vergata”, 00166 Rome, Italy; 4Department of Biomedicine and Prevention, University of Rome “Tor Vergata”, 00166 Rome, Italy; palombi@uniroma2.it

**Keywords:** COVID-19, time series analysis, emergency medical services, public health, prediction models

## Abstract

(1) Background: During the COVID-19 outbreak in the Lazio region, a surge in emergency medical service (EMS) calls has been observed. The objective of present study is to investigate if there is any correlation between the variation in numbers of daily EMS calls, and the short-term evolution of the epidemic wave. (2) Methods: Data from the COVID-19 outbreak has been retrieved in order to draw the epidemic curve in the Lazio region. Data from EMS calls has been used in order to determine Excess of Calls (ExCa) in the 2020–2021 years, compared to the year 2019 (baseline). Multiple linear regression models have been run between ExCa and the first-order derivative (D’) of the epidemic wave in time, each regression model anticipating the epidemic progression (up to 14 days), in order to probe a correlation between the variables. (3) Results: EMS calls variation from baseline is correlated with the slope of the curve of ICU admissions, with the most fitting value found at 7 days (R^2^ 0.33, *p* < 0.001). (4) Conclusions: EMS calls deviation from baseline allows public health services to predict short-term epidemic trends in COVID-19 outbreaks, and can be used as validation of current data, or as an independent estimator of future trends.

## 1. Introduction

### 1.1. Background and Rationale

#### 1.1.1. Health Emergency Systems in Italy and ARES 118 in the Lazio Region

Italian National Health System is the only authority tasked with responding to medical rescue and medical health emergency calls in territorial (i.e., non-hospital) settings. This means that all emergency ambulance services, even if belonging to private companies, are framed within a single public service, whose governance may be either local or regional, according to national and regional regulations [1]. In Lazio, a region counting over 6 million inhabitants, the Emergency Medical System (EMS) has been managed by the “Azienda Regionale Emergenza Sanitaria 118” (ARES–Regional Health Emergency Agency) since 2004 [2,3]. “118” is the toll-free emergency telephone number for emergency medical service in Italy, even if since 2017 it has also been possible to call by the toll-free European emergency number 112, although this is a general police/fire/medical number; 118 is present in the name of other regional public EMS in Italy as well [4,5].

Staffing over 900 health professionals and several other technical and administrative figures in 2020, ARES 118 responds to Health Emergency via its own ambulances fleet network (either proprietary, or in accord with private local ambulance companies or volunteer organizations), its Helicopter Emergency Medical Service (HEMS) running over 3 helicopter bases, ant its 3 Operative Centrals (OC), whose territorial competence is North Lazio, South Lazio, and Metropolitan City of Rome, respectively. All Medical Emergency calls are addressed to either of these OC. Further units are responsible for specific emergency responses (such as neonatal emergency, maxi-emergencies, blood and organ delivery, and others). ARES manages millions of emergency calls each year, and hundreds of thousands of ambulance interventions [6].

#### 1.1.2. COVID-19 Outbreak in the Lazio Region during Years 2020–2021

The COVID-19 outbreak in Italy is believed have begun on 30 January, 2020, with the arrival of two infected tourists from China; these two cases were also the first ones reported in Italy, although hints have been found that the disease was already circulating during the previous months [7]. At any rate, the epidemic waves affected the Lazio region just like the rest of Italy, and posed a heavy burden on the local health system [8]. Several managerial strategies had to be adopted in order to mitigate the epidemic’s effect, such as the adoption of a “Hub and Spoke” model for COVID structures, rescheduling of elective medical procedures, and other public health measures [9]. Additionally, knowledge, risk perception and practice to prevent its spread varied among the population [10]. At the time of writing (April 2022), over 1,400,000 COVID cases were reported in the region, and over 10,000 deaths [11,12].

#### 1.1.3. COVID-19 Outbreaks and EMS Calls

Worldwide, it has been acknowledged that the COVID-19 outbreak can lead to a variation in EMS demand, and a strong correlation between EMS calls and hospitalization pattern has already been identified, with a rate of (EMS transportations):(hospital admissions) of 2.5:1 [13]. Moreover, both EMS calls and epidemic trend were reported to be changing in response to social restriction strategies that were enacted on national or local level [14]. This effect was seen both on the magnitude of the calls, and on the nature of the patients’ demands [15]. Such result, that were all found during the initial phases of the pandemic, all called for innovative strategies in both emergency planning and outbreak response, in order to better navigate across the kind of difficulties that Health Systems faced during the pandemic period, and that were strongly felt by operators, researchers and patients alike [16,17,18,19]. Additionally, while several attempts are made focused on predicting EMS demand [20], there is no established knowledge on the role of EMS calls analysis in forecasting an ongoing epidemic trend.

### 1.2. Objectives

The objective of the present study is to investigate if there is any correlation between the variation in numbers of daily calls to health emergency services during the pandemic outbreak (compared to baseline values of 2019), and the short-term evolution of the epidemic waves in the years 2020–2021.

## 2. Methods

### 2.1. Study Design

Quasi-experimental ecological study, with Interrupted Time Series Analysis [21] on routinely collected data.

RE.C.O.R.D. Guidelines have been used for reporting the study results [22].

### 2.2. Setting and Population

The study was conducted in the Lazio region in Italy, an area with over 6 million inhabitants [3]. All of the resident population was potentially susceptible for analysis. All EMS calls data during the observation period were included in the analysis.

### 2.3. Variables and Data Sources

The following variables have been used for data analysis and modelling:Daily COVID-19 incidence;Daily COVID-19 admissions;Daily COVID-19 admissions in Intensive Care Units (ICU);Daily deaths among COVID-19 patients;Daily toll-free calls to EMS.

Data (1)–(4) are of public domain, and are provided by the Italian Ministry of Health [11]; data (5) are routinely collected by *ARES* for its institutional activity, and are property of *ARES*.

All data were natively anonymous. Analyzed timespan ranged from 1 January 2019 to 1 February 2022.

### 2.4. Statistical Analysis

Microsoft^®^ Excel^®^ v.2016 MSO was used for calculation and graphing. All data were analyzed at the Department of Biomedicine and Prevention of the University of Rome “Tor Vergata”. Each Data set was considered a Time Series of y data, ordered in function of the time t. k–period moving average (MA) of such variables, $\left(k\right){\mathrm{MA}}_{y}$, was as such defined [23]:(1)$$\left(k\right){\mathrm{MA}}_{y}=\left\{\begin{array}{c} \frac{1}{t}\overset{t}{\underset{i=1}{\sum }}y_{i},\;\;\;\;\;\;\;\;\;0<t<k\\ \frac{1}{k}\overset{t}{\underset{i=t-k+1}{\sum }}y_{i},\;\;\;\;\;\;t\geq k \end{array}\right.$$

#### 2.4.1. Epidemic Trend

Variables (1)–(4) were computed and compared using Pearson’s ρ [24] in order to test for co-linearity and validating the usage of any of them as a proxy of epidemic trend in any successive analysis. Once such co-linearity was proved, ICU admissions number was considered the optimal proxy, since it cannot be altered by circumstantial factors (such as change in case definition, change in case testing strategy, or change in case testing effectiveness).

For Variable (3) (Daily COVID ICU admissions), for each data point $y_{t}$, the value of the first-order derivative ($D\prime_{ICU}$) of its $\left(7\right)MA$, $\dot{y_{t}}$, has been calculated.

The first-order derivative of a curve at a point  y is the slope (angular coefficient m of the tangent line) at that point. Since no simple function could satisfactorily approximate the real observed epidemic curve, instead of using an indirect method (by performing a spline interpolation of the curve, and then computing the first order derivative of each equation) a direct method was used by approximating $\dot{y_{t}}$ to the first-order derivative of its two days moving average:(2)$$\dot{y_{t}}≅\;\dot{\left(2\right)MA_{y}}$$
where $\dot{\left(2\right)MA_{y}}$ can be approximated to the angular coefficient m of the line passing by two consecutive values of $\left(2\right)MA_{y}$ ordered in function of time t, according to the following formula [25]:(3)$$m=\frac{\left(2\right)MA_{y}{}_{2}-\left(2\right)MA_{y}{}_{1}}{t_{2}-t_{1}}$$

Since $t_{1}$ and $t_{2}$ are two consecutive days, $t_{2}-t_{1}=1$; therefore
(4)$$m=\left(2\right)MA_{y}{}_{2}-\left(2\right)MA_{y}{}_{1}≅\dot{\left(2\right)MA_{y}}≅\;\dot{y_{t}}$$

#### 2.4.2. Trend of EMS Calls

Data on number of daily EMS calls have been adjusted for weekly seasonality, using the method of Classical Multiplicative Decomposition [26]. Series relative to the years 2019, 2020 and 2021 were directly compared by subtracting the respective value in corresponding dates, hence obtaining an estimation, from 2020 onwards, of the excess (or defect) in the number of daily EMS calls (ΔCalls) from baseline (2019) number of daily EMS calls. This allows us to directly compare trends observed in the same date of different years, unaffected by the variability due to being a different day of the week (like Wednesday vs. Sunday). This method, unfortunately, does not allow correction for the effect of fixed or mobile occurrences (such as Christmas, Easter, New Year’s Day, local or national festivities). However, since long-term trend is extrapolated, it is possible to observe the effects of casual or cyclical causality.

#### 2.4.3. Regression Analysis

Eleven univariate regression analyses between ΔCalls and Epidemic trend derivative $\dot{y_{t}}$ for ICU admissions ($D\prime_{ICU}$) at 0, 5, 6, 7, 8, 9, 10, 11, 12, 13 and 14 days, respectively, have been performed in order to show the independent predictive power of ΔCalls, during an outbreak, in forecasting the evolution of the outbreak itself on different time intervals.

Linear regression is written in the form of
(5)$$\hat{y}=\beta x+\varepsilon$$
where $\hat{y}$ is the expected value of the dependent variable, *x* is the independent variable value, β is the regression coefficient, and ε the intercept value or error term [27].

Typically, regression analysis involves calculation of the R-Squared measure (R^2^). R^2^ is a statistical measure that represents the proportion of the variance for a dependent variable explained by the independent variable. Whereas correlation explains the strength of the relationship between an independent and dependent variable, R^2^ explains to what extent the variance of one variable explains the variance of the second variable. Therefore, if the R^2^ of a model is 0.50, then approximately half of the observed variation can be explained by the model’s inputs [28].

#### 2.4.4. Ex-Post Data Correction

Due to a hardware failure in the ARES Data warehouse, it has been observed a systematic data loss on EMS calls in the month of May 2019. Data has been manually corrected, by adding a flat +500 units for each day since 1 to 30 May 2019.

#### 2.4.5. Validation

Since data from different settings and population is not available, external validation was not viable.

Internal validation was performed using the k-fold cross validation method [29]. Four-fold cross validation was performed by splitting data from 13 February 2020 to 31 December 2021 into 4 stacks of 172 observations each and leaving one stack out for each prediction iteration. R^2^ values found for each iteration were then averaged and confronted with actual regression results.

#### 2.4.6. Post Hoc Analysis: Forecasting Power

Given the positive correlation found between ΔCalls and $D\prime_{ICU}$ on a 7-day lag, a fast forecasting function has been empirically proposed.

In order to build the function, the following tenets were considered:(A)It must have current $D\prime_{ICU}$ value as starting value for estimation (imput);(B)It must be a function of current ΔCalls fluctuation;(C)The two variables must be linked by the proposed regression model.(D)Its output would be the expected value of $D\prime_{ICU}$ after 7 days,$\;{D^{\prime }}_{ICU\left(expected\right)}$.

The proposed forecasting function has been built by transforming the ΔCalls values in their respective z-scores (z), in order to make the variable independent from its measure scale. Function has then been empirically written as: (6)$${D^{\prime }}_{ICU\left(expected\right)}={\left(zD\prime_{ICU}\right)}^{\left(1-R^{2}\right)}$$

### 2.5. Privacy, Ethical Committee Approval and Informed Consent

This study was performed on data that are either in public domain, or routinely collected for administrative purpose. It did not require any patient to sign an informed consent, as per authorization of the National Italian Privacy Board (Garante per la Protezione dei Dati Personali) n° 9/2016 [30].

All data were natively extracted anonymously.

No ethical approval was required nor sought, due to the study being retrospectively conducted on routinely collected\public domain data.

## 3. Results

### 3.1. Participants

From 1 January 2019 to 31 December 2021, a grand total of 1,926,117 EMS calls have been observed in the Lazio Region, resulting in 1,257,310 Rescue Missions.

From 23 February 2020 to 31 December 2021, 787,688 people were admitted to hospitals with COVID-19 symptoms, and 91,708 people were admitted to ICUs while positive for COVID-19 (10.4%). Additionally, 505,273 people were tested positive to COVID-19 in the same period, and 9269 people died while positive for COVID-19.

### 3.2. Epidemic Trend

The epidemic trend in Lazio followed the Italian national trend, with an exponential behavior (1st wave) since February 2020 until April 2020. After a short plateau, COVID-19 incidence started decreasing in the following months due to the generalized lockdown. Incidence remained stable at low values during summer 2020, but an exponential trend started again since the end of August 2020 with a peak during November 2020 (2nd wave). After the beginning of the vaccination campaign, daily COVID-19 incidence decreased again, and came back to low levels in summer 2021 after experiencing a 3rd wave, shortly after a major relaxation in restriction policies after the first vaccination tranche on frail people and health personnel. A new increment, of lesser volume, happened after August 2021 due to the diffusion of exotic COVID-19 variants, of which the vaccination protection provided by available vaccines was sub-optimal (4th wave). Incidence began to raise again at the end of year 2021. Figure 1 depicts the epidemic trend as described.

Symptomatic admissions and ICU admissions showed a trend coherent with the epidemics one, and kept being roughly the same fraction of the diagnosed cases at least until higher vaccination percentage had been reached among population. Figure 2 shows ICU admissions, hospital admissions and COVID-19 incidence on a log scale.

Pearson’s ρ values, calculated for each couple of the epidemiological Variables (1)–(4), show a strong collinearity as expected. Results are shown in Table 1.

### 3.3. Trend of EMS Calls

EMS calls in Lazio region are not homogenous in the studied period. On January and February 2020, the trend was basically the same as in the same months of 2019; prior to the pandemic outbreak, there was a surge in EMS calls, then there was a drastic and sustained diminution in EMS calls during lockdown. In summer 2020, the EMS call trend was again back to 2019 levels, and remained similar to the baseline trend until October 2020: at that point, a new surge in EMS calls was observed, followed by the 2nd wave of epidemic outbreak. This pattern was also observed, although in lower magnitude, during 2021 (Figure 3).

It is widely known that not all EMS calls result in the patients’ transportation to a hospital facility. On average, it has been observed that the calls/transportation ratio was ≈ 1.5, but the value itself has shown important fluctuation in coincidence with the first two waves of epidemic progression (Figure 4). No obvious pattern has been observed afterwards.

Difference in calls number (ΔCalls) in 2020 and 2021 compared to baseline (2019) has been calculated and plotted against the first order derivative D’ of ICU admissions after 7 days ($D\prime_{ICU}$). The two curves show a very similar behavior (Figure 5), tracking closely until Spring of 2021. Afterwards, their relation is still constant in regard to the direction, but the magnitude of the effect appears less evident at eye-glance. This apparent discrepancy is due both to the difference in the order of magnitude of the two curves, and in their respective meaning: ΔCalls represents a mere number subtraction, being the difference in raw calls from previous year; $D\prime_{ICU}$ represents the slope of ICU curve admissions, and it is more susceptible to rapid vertical change if the starting numbers are small. This means that comparative small variation in raw numbers show a higher variation in the curve slope, a phenomenon showing the same behavior in other epidemiological parameters, such as the reproduction number R [31].

### 3.4. Regression Analysis

Univariate regression analysis between ΔCalls and Epidemic trend derivative $\dot{y_{t}}$ for ICU admissions at 0, 5, 6, 7, 8, 9, 10, 11, 12, 13 and 14 days, respectively, have been performed, and results are reported in Table 2. More details on the regression models themselves, along with raw data and calculations, are provided in Appendix A.

### 3.5. Validation

Internal validation results are shown in Table 3. Data included was relative to the period 17 February 2020–31 December 2021. The 684 observations were split into four consecutive stacks. Each stack was consecutively taken out of the analysis and then reintroduced as the next one was removed, in four steps. Mean of R2 results was then calculated and confronted with actual R^2^ for the entire observations set.

### 3.6. Forecasting Power

Using Equation (VI), a daily forecasting has been proposed in order to obtain, from current $D\prime_{ICU}$ value and current ΔCalls fluctuation, the expected $D\prime_{ICU}$ value after 7 days. Results are graphically shown in Figure 6. Its performance was tested using a linear regression model between expected and actual $D\prime_{ICU}$ values. R^2^ for the estimation on the whole 2-years period suggested a mildly moderate power (Adjusted R^2^ = 0,38, *p* < 0.001). However, when the same estimation is performed only on 2020 data, it show a much better performance (Adjusted R^2^ = 0,56, *p* < 0.001) A detailed interpretation of this result is proposed in the Discussion sections.

## 4. Discussion

### 4.1. Key Results

Prediction of need for hospitalization during epidemic outbreak is a staple in any containment strategy for infectious diseases. Currently, the analysis of excess of EMS calls seems to be a candidate predictor in estimating future need of hospitalization in the population. A value of R^2^ of 0.33 between the slope of the epidemiological curve and the excess of EMS calls indicates a moderate correlation between the two variables (33% of variation in one variable can be explained by the variation of the other), meaning that EMS calls excess is an estimator of 7-days ICU admissions trend progression. Since ICU admissions are a constant fraction of hospital admissions (correlation ≈ 97%), EMS calls excess is also an estimator of future admissions, and can be used as an indicator of short-term stress increase/decrease on the health system as a whole during an epidemic outbreak.

A side result of this research was the observation that the rate between EMS calls and EMS interventions was not constant over time. It has been observed that, while the rate value was globally bouncing around 1.5, it was much higher at the beginning of the pandemic waves, consistently with the raise in raw EMS calls number. This means that many calls made at the onset of the epidemic waves did not result in an EMS intervention, hinting that many people may have sought EMS support in a likely inappropriate fashion. It is possible that such behavior is reinforced by concurrent social distancing measure, with the side-effects of exacerbating difficulties among more frailty sector of the population, who sought public EMS service as a possible solution to more structural problems [32]. Such behavior is unfortunately well known especially among the lonely and the elderly, and integrated care approaches should be a focus of health systems in the near future [33,34].

### 4.2. Limitations

This study has two major limitations:It has been partaken during a time when the COVID-19 pandemic was the major epidemic outbreak on the population. Lag time on ICU admissions depends on the epidemic characteristics of the virus itself: a pathogen with different epidemiological characteristics (such as serial interval, virulence, and infection rate) may yield to different results in terms of lag from EMS calls excess and actual increase in hospitalization. This of course does not affect the validity of current results or methodology, since the main purpose of current work was to investigate if EMS calls trend analysis is a viable tool in infectious diseases control, and if it can be a valid predictor in the right conditions.This study used only 2019 as a reference year for baseline drawing. Ideally, data from several “baseline” years should be used (for instance, European Mortality Monitoring EuroMOMO uses data from the previous 5 years in order to estimate excess mortality) in order to have a better baseline modelling [35]. Unfortunately, this limitation could not be overcome due to lack of an adequate data warehouse for *ARES* activity before 2019.

An interesting consideration is due in regards to the effect of the vaccination campaign on both the epidemic itself, and the proposed model validity. While any inference on the efficacy of the adopted vaccination strategy is out of the scope of the present work, it can be observed how COVID-19 incidence and ICU admissions dropped immediately after the beginning of the vaccination campaign, and while the epidemic case raw numbers saw a rise at the end of 2022, ICU admission rate was comparably lower. This is more evident in Figure 2. While the distance among all curves is constant up until the rolling on of the vaccination campaign (April–May 2021), the pure COVID-19 incidence (positive tested people) then rises above ICU and death curves, suggesting the diffusion of less lethal COVID-19 variants, or an overall protective effect of COVID-19 vaccination against most extreme adverse events in case of actual contagion. Given current literature findings on the matter, the second option is considered the most likely by the authors, although the two scenarios are not mutually exclusive from a theoretical standpoint [36].

It has been observed that EMS calls forecast the ICU admission trend more than the actual number of people tested positive; given the mentioned considerations, this does not come as a surprise. Actually, it should be considered as a point of interest in this research, since COVID clinical presentation has been switching in the past few months, and the main focus (from a public health perspective) is still on its impact on health structures and systems, whose resilience has been heavily tested in the last two years [37,38]. A growing amount of positive, yet asymptomatic or mildly symptomatic cases, is of little interest when compared to the warnings of a possible imminent surge in ICU demand: in such scenarios, EMS calls analysis, alone or in combination with other indicators, can be useful in health programming on a short-time period, and on a local level. This is even more cogent when confronted with all the amount of information that has been circulated on COVID-19, on both scientific circles and social media, but has not been supported by adequate evidence, either for lack of data, or methodological flaws [39,40].

As for the forecast power demonstrated by the proposed modeling method, it should be noted that its equation is strongly dependent by the variation in ΔCalls. As previously stated, this parameter should be estimated using multiple reference years, a strategy that was unfortunately not available for this research: the authors had to estimate it using only 2019 as reference year. This means that random mid-term events that have could have locally modified the EMS calls value in 2019 may lead to a spurious ΔCalls estimation for successive years. It should be noted that actual estimation of the variables begins on 14 February 2020 (there is no current ICU admissions data prior to that date in the model, hence the flat curve in the first months), and that the discordance between expected and observed values appear mostly in the first few months of 2021. We believe that to be a consequence of a potentially biased estimation of the ΔCalls parameter relative to that period, however there is no possibility of checking for that given no prior reference year.

In that regard, Internal Validation Analysis results support this conclusion. The whole model was much more fit to the data when the values from the 1st part of 2021 were excluded from the analysis. Unfortunately, values from the first 45 days of 2020 are invalid (no epidemics at the time), so there is no possibility of confrontation between the two years. This is unfortunate, since, as seen in Figure 4, the two curves of ΔCalls and $D\prime_{ICU}$ are analogous for many tracts, but the discrepancy observed in the initial period of 2021 drags down many of the respective correlation values. Such discrepancy again seems to disappear in the second half of 2021.

## 5. Conclusions

### 5.1. Interpretation

Due to the obvious limitation of getting a reliable epidemic estimator of COVID-19 incidence, we chose to base the description of the epidemic trend using ICU admissions and COVID-19 deaths as proxies of actual disease incidence. Actually, the fact that, among immunological naive populations, COVID-19 Hospital\ICU admissions and deaths are a function of COVID-19 incidence, and that the rate is constant in time, had been already ascertained during the first phases of the pandemic in 2020 [41]. Our data and our collinearity analysis are coherent with current literature findings. Of course, this assumption may not hold in the case of higher vaccine coverage among a population: hence, the importance of understanding and intervention in cases of vaccine hesitancy in both adult and pediatric populations [42,43,44].

Discrepancy between admission and positive cases, with the apparent paradox of symptomatic admissions being more than positive cases, may be due to several factors. The authors believe that several elements may be taken into account in explaining this phenomenon:Presence of a proportion of false-negatives;Limitation on the number of available tests, especially during the “first wave” phase, with a hard upper threshold on daily positive cases that could be identified;Late adoption of quick testing strategy as a viable case definition;Variability in symptoms onset and swab positivization, leading to disease underreporting.

As a matter of fact, seroprevalence studies conducted in Italy have shown that COVID-19 infection cases were higher than those actually diagnosed. In the Lazio region a seroprevalence of at least 1% after the first two waves of 2020 was reported [45]: this would mean over half a million cases during the first year of epidemic only. By contrast, only 163.050 positive cases were recorded at the time.

The R^2^ value that has been reported in this paper is relative to the whole 2020–2021 period. If needed, a hierarchical model could be proposed, in order to break down the strength of the prediction in different phases of the pandemic course. Of course, the more data is added, the less uncertainty is found in the final results, and just as is the case with hierarchical models, excessive focus on some specific portion of the curves could lead to model overfitting [46].

### 5.2. Generalizability

Not only does this study data confirm what is known on COVID-19 epidemics in terms of incidence and rate of hospital\ICU admissions and death rate, it also provide a useful short-term provisional tool for territorial services and public health authorities. By comparing current EMS calls trend with baseline trend, it is possible to forecast the shape of the epidemiological curve, with no need of actively monitoring current incidence. This can be extremely useful for any decisors level, since it allows a more data-driven resource allocation strategy. It also allows implementing a more proactive approach to disease control, using a prospective indicator such as EMS calls excess. This can be a crucial tool in scenarios that are still based on retrospective or “reactive” indicators, such as current incidence or current ICU occupation rate. For example, since the magnitude of EMS calls excess contributes in forecasting the shape of the local epidemiological curve 7 days later, it is possible to provide adequate support to local structure (i.e., programming an expansion or a contraction of ICU beds availability) at least a week in advance.

A similar approach has already been implemented in other locations such as New Mexico, where predictive current incidence-based models were used [47]. Additionally, a study by Rivieccio et al. [48], showed that EMS calls foreshadow an epidemiological trend; however, the methodology used for this analysis was different from ours, as it used incidence as main prediction outcome, crude EMS calls value as predictor, wavelet analysis and Monte Carlo simulation for lag estimation. Such methods are not of standard use in Public Health practice, and usually require specialized personnel and dedicated software. To our knowledge, our study is the only one that relates a baseline deviation trend in EMS calls to a variation of the slope of the epidemiological curve, using simple and well-established methodologies such as trend decomposition, excess estimation, and univariate regression analysis. Regardless of the adopted method, the results from Rivieccio et al. pointed towards a lag between epidemic progression and incoming EMS calls peak of 6 days, with a wide confidence interval (13, −1). Our results seem to confirm their initial intuition, since the most fitting curve overlaps were seen between the 6th and the 9th day, adding further evidence to what is already known. An additional strength of this work is the provision of a simple empirical equation for trend forecasting, whose performance, albeit not exceptional, is still in line with current findings.

This study was performed on COVID-19 epidemics, using data from 2 years and from a potential population of almost 6 million. Its principles, however, are valid for any epidemic outbreak, and can be a useful weapon in the (unfortunately still limited) arsenal of Disease Control centers, especially on local level [49]. Not only can EMS calls deviation from baseline be used for forecasting purposes, but it can also be used as an independent validation technique for current or past data–for instance, allowing estimation of underreported cases among the population, or allowing confrontation with current R_t_ values.

The authors believe that the proposed method can be used in any local area with a toll-free EMS call number, since it does not require any prior inference of unknown parameters relative to the population setting. The only thing that should be variable would be the lag value, in relation to the epidemic characteristics of the involved pathogen. Validation studies on different setting and populations could add even more evidence on the strength of the proposed method.

## Figures and Tables

**Figure 1 ijerph-19-05951-f001:**
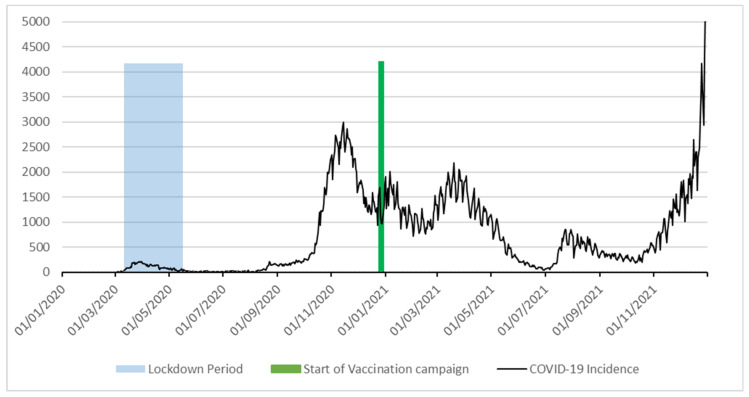
COVID-19 incidence in Lazio region, Italy, 2020–2021.

**Figure 2 ijerph-19-05951-f002:**
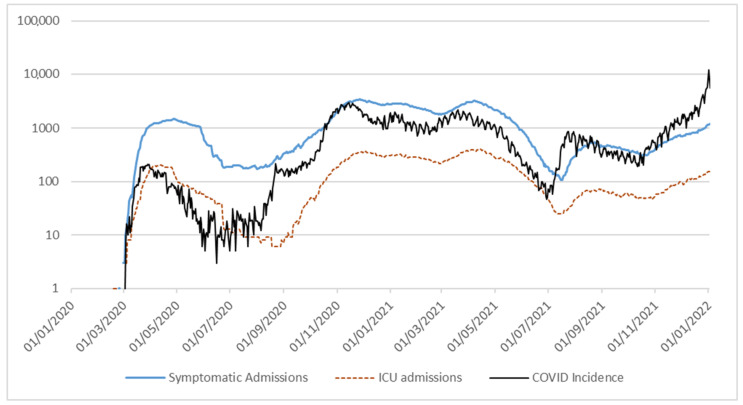
Hospital and ICU admissions, and COVID-19 incidence in Lazio region, Italy, 2020–2021 (log scale).

**Figure 3 ijerph-19-05951-f003:**
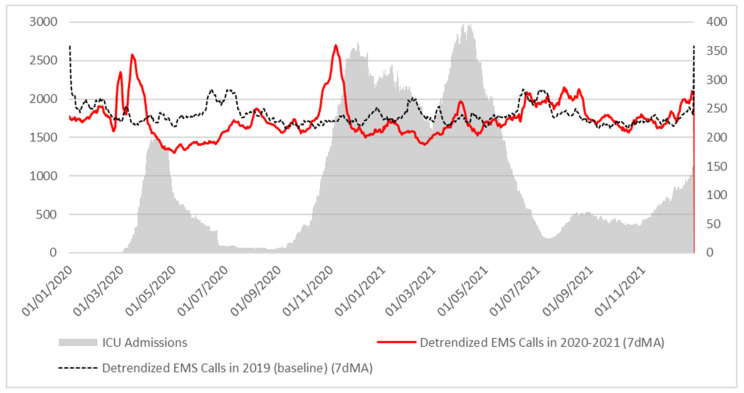
EMS calls and ICU admissions in Lazio region, Italy, 2020–2021.

**Figure 4 ijerph-19-05951-f004:**
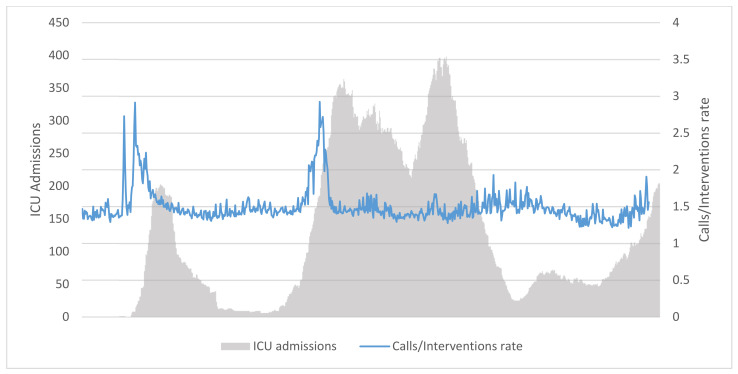
Calls/Interventions rate and ICU admissions in Lazio region, Italy, 2020–2021.

**Figure 5 ijerph-19-05951-f005:**
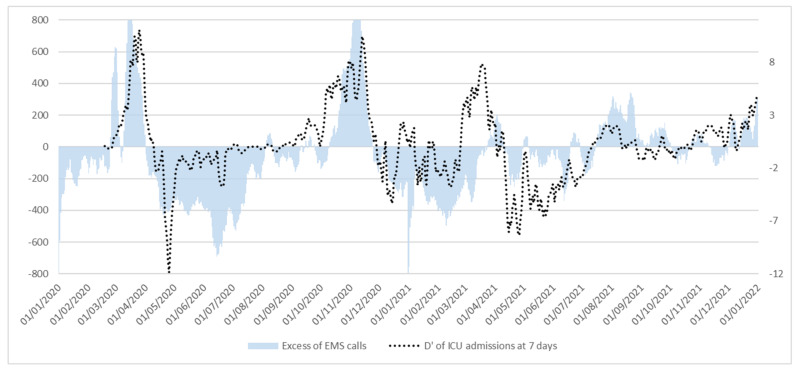
Excess of EMS calls (top) and 1st-order derivative of ICU admission at 7 days (bottom) in Lazio region, Italy, 2020–2021.

**Figure 6 ijerph-19-05951-f006:**
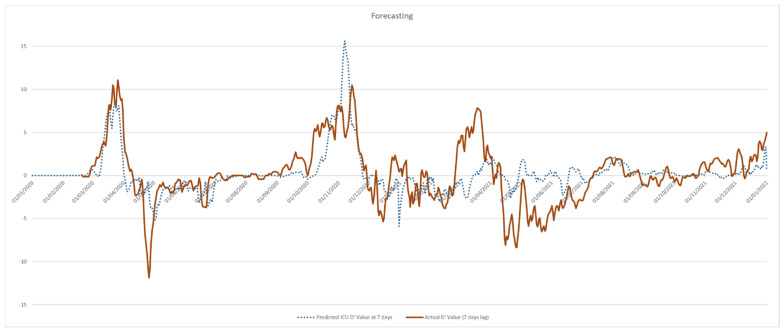
Predicted vs. expected values of first order derivative of 7-days ICU admissions in Lazio Region, Italy, 2020–2021.

**Table 1 ijerph-19-05951-t001:** Collinearity of epidemiological variables.

	Symptomatic Admissions	ICU Admissions	Total Admissions	COVID Incidence (14 Days Prior)	Daily Deaths
Symptomatic Admissions	1				
ICU admissions	0.968823	1			
Total admissions	0.999663	0.974927	1		
COVID Incidence (14 days prior)	0.841558	0.837922	0.843642	1	
Daily deaths	0.863062	0.835356	0.862688	0.760818	1

**Table 2 ijerph-19-05951-t002:** R^2^ and *p*-values for univariate regression models between ΔCalls and first order derivative of ICU admissions in Lazio Region, Italy, 2020–2021. Most informative value results are in bold. ICU: Intensive Care Unit.

1st Derivative of ICU Admissions at	Adjusted R^2^ ΔCalls	F-Statistic *p*-Value
0 days	0.247	2.2259 × 10^−43^
5 days	0.3191	4.6315 × 10^−58^
6 days	0.326	1.5130 × 10^−59^
7 days	**0.328**	**5.6168** × **10^−60^**
8 days	0.3278	6.156 × 10^−60^
9 days	0.3257	1.8340 × 10^−59^
10 days	0.3191	4.7653 × 10^−58^
11 days	0.3095	5.099 × 10^−56^
12 days	0.296	3.3879 × 10^−53^
13 days	0.2799	6.9579 × 10^−50^
14 days	0.2627	1.8906 × 10^−46^

**Table 3 ijerph-19-05951-t003:** Four-fold cross validation of 7-days lag model.

Observations Set	R^2^ Values
Fold 1	0.24717
Fold 2	0.244523
Fold 3	0.481349
Fold 4	0.32992
**Mean**	**0.32574**
**Global R^2^ estimate**	**0.321919**
**Difference**	**−0.00382**

## Data Availability

All relevant data has been reported in the paper or is available as Appendix A.

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
