# Peer review of "Emergency Medical Services Calls Analysis for Trend Prediction during Epidemic Outbreaks: Interrupted Time Series Analysis on 2020–2021 COVID-19 Epidemic in Lazio, Italy"

_ijerph, 2022, doi:10.3390/ijerph19105951_

Round 1

Reviewer 1 Report

Thank you for allowing me to review this manuscript. This manuscript entitled "Emergency Medical Services Calls Analysis for Trend Prediction during Epidemic Outbreaks: Interrupted Time Series Analysis on 2020-2021 COVID-19 Epidemic in Lazio, Italy", aims to investigate if there is any correlation between the variation of 68 in number of daily calls to the Health Emergency Service during the pandemic outbreak 69 (compared with baseline values ​​in 2019), and the short-term evolution of the epidemic waves 70 in the years 2020-2021.

It is an interesting and highly relevant article today, although it has several limitations that make it suitable for publication in this journal. These limitations are detailed below:

- In the introduction, it would be important to justify in more detail the importance and timeliness of the topic of study and add more citations.

- In the material and methods section it is described in detail. However, as a signal and weakness, we can highlight that different important ethical considerations are not reflected. It would be necessary to indicate if the study has the authorization of the ethics committee.

- The results are presented in a clear and orderly manner. Also, an interpretation of them is reflected. However, in the tables the acronyms used in them are not specified at the foot of the table.

- The discussion is a fundamental section. However, in the manuscript it is very brief. It would be necessary to expand this section and add studies that can discuss and compare the results obtained in this manuscript.

- In relation to the conclusions, they are clear and precise. However, we recommend indicating more precisely the implication in clinical practice of the same.

nice job

Author Response

Many thanks for overseeing a timely review of our paper.

We are very grateful for the constructive comments from the reviewers; even more so, since the more precise hindsight requested brought us to a re-analysis of the data. In this process, we recognized that one of our main results (the optimal value of 10-days lag in predictivity) was wrong due to a systematic calculation error, and lead us to a slightly wrong conclusion. We have updated the entire paper, whose staple structure remained nonetheless the same since the rest of the results were not affected by this error. We recognized that 7-days lag is actual the optimal fitting vale for prediction, and the retrieved R2 is lower than originally suggested.

In order to add strength to the proposed evidence, Internal Validation analysis has been performed.

While being sorry for this inconvenient, we remark that the main conclusions have not been deeply affected.

Also, as per Rev. 2 suggestion, we performed a forecast analysis using the retrieved model, showing expected and observed results on a 7-days lag basis.

All data, analysis and models have been updated in supplementary materials as well.

We include here point-by-point responses to the reviewers’ suggestions.

Many thanks and best regards, the authors.

Rev. 1:

1. In the introduction, it would be important to justify in more detail the importance and timeliness of the topic of study and add more citations.

Introduction has been updated with more information on the investigated topic, more literature references have been inserted, and a new section "1.1.3. COVID-19 outbreaks and EMS calls" has been added, also as per Rev. 2 suggestion.

2. In the material and methods section it is described in detail. However, as a signal and weakness, we can highlight that different important ethical considerations are not reflected. It would be necessary to indicate if the study has the authorization of the ethics committee.

Thank you for the heads up. Actually, this study has not been authorized by any ethical committee: being an observational, ex-post study conducted on routinely collected data, it does not require any formal authorization. In order to better explain this, subsection "2.5. Privacy, Ethical Committee approval and informed consent" has been added in the manuscript, with legislation references.

3. The results are presented in a clear and orderly manner. Also, an interpretation of them is reflected. However, in the tables the acronyms used in them are not specified at the foot of the table.

Thank you for the positive feedback. Acronyms have been added in tables as suggested.

4. The discussion is a fundamental section. However, in the manuscript it is very brief. It would be necessary to expand this section and add studies that can discuss and compare the results obtained in this manuscript.

Results and Discussion sections have been expanded with further comments and some insights on points that, despite originally out of the scope of present research, were nonetheless relevant, as per Rev. 2 suggestions also. Moreover, our results have been compared with the only work we know of that, up to date, is similar enough for a meaningful comparison.

5. In relation to the conclusions, they are clear and precise. However, we recommend indicating more precisely the implication in clinical practice of the same.

Thank you for the positive comment. The section has been further expanded and some practical implication have been suggested.

Reviewer 2 Report

Review of “Emergency Medical Services Calls Analysis for Trend Predic-2 tion During Epidemic Outbreaks: Interrupted Time Series 3 Analysis on 2020-2021 COVID-19 Epidemic in Lazio, Italy”

Overview: This paper seeks to examine the relationship between the COVID-19 pandemic on EMS calls in Lazio Italy.

Abstract:

  1. The results section in the abstract is very short and provides no actual reporting of raw results.

Background:

  1. The background section is very general and focuses on describing the Italian context. The authors should focus on existing literature examining relationships between COVID-19 (or other outbreaks) and their relationship to EMS calls.

Analyses/Results:

  1. The final analysis model could use clarity. Please confirm that the epidemiological variables assessed for collinearity (see Table 1) resulted in the final selection of using only the ICU admissions variable in the model. I was not sure if the other variables were dropped in the Methods description.
  2. Furthermore – perhaps this needs a specific statistical review, but I would assume the authors could use time-series analyses utilizing a regression model versus direct calculation in Excel. There are a lot of equations described in the text although whether this is a valid analysis approach eludes me.
  3. The selection of ICU admission seems to be the appropriate variable used to quantify the epidemic’s trend. This was supported by the collinearity results reported in Table 1.
  4. This study adds to evidence that EMS calls are predictive of ICU admissions during the COVID-19 pandemic, particularly prior to wide availability of COVID-19 vaccines. Figure 4 is striking in demonstrating how a lag of 10 days between EMS calls and the trend in ICU admissions are analogous. The authors acknowledge how these findings are specific to this epidemic, and these findings may change depending on characteristics of the pathogen (e.g., virulence). However, the authors could comment on how availability of the vaccine may affect the predictive value of EMS calls associated with ICU admissions.
  5. In Figure 4, change in calls and D’ of ICU admissions tracks closely until Spring of 2021. For the months thereafter, what do you make of why EMS calls decreased while ICU admissions were notably increased. The patterns do not seem to track as strongly after this date. Could you explain or talk about this discrepancy?
  6. Table 2 show the results of the univariate regression models between change in calls and epidemic trend. Using 10 days as the derivative of ICU admissions looks well supported, but could you break this into 2020 and 2021 along with what you report as combined results for 2020-2021? Would the derivative day still be the same in both years?

Discussion:

  1. A strength of this study is that it draws from a large sample of EMS calls in a populous region of Italy. EMS calls are centralized within one management system, ARES. It could be noted that not all calls to EMS result in action taken by EMS, either through responder contact or through transport to a medical facility. What proportion of calls to EMS are transported and did that change throughout the study period? It is likely beyond the scope of this study, but it could be interesting to see if there was a correlation between certain symptoms (e.g., respiratory distress) that were the reason for the EMS call and ICU admissions.
  2. The discussion section is very short (one paragraph) and does not situate the authors findings in the existing literature on this topic. How do the authors findings compare to existing literature utilizing either EMS calls as a predictor of a future epidemic progression – or for other diseases / issues?
  3. Could the authors examine whether their model and approach either a) extends to other countries or b) whether it can actually predict forward in the pandemic? The pandemic is ongoing, so the authors could make a prediction using their model in this context and then validate their model by comparing it to the actual trend after prediction.

Minor Points:

  1. Check for spelling errors throughout the manuscript. For example, “spleen” interpolation (line 112) should probably be spline interpolation. Line 74 “Quasi-sperimental” is likely Quasi-experimental?
  2. Confirm population estimate of Lazio region. It says over 5.5 million on line 40 and over 6 million on line 78-79, and almost 6 million on line 281.
  3. HES was not defined. I assume this is Health Emergency Services on line 69. Consider calling them emergency calls or calls. “EMS calls” is also used, see line 123. There are many abbreviations and terms using in reference to calls in this paper.

Author Response

Many thanks for overseeing a timely review of our paper.

We are very grateful for the constructive comments from the reviewers; even more so, since the more precise hindsight requested brought us to a re-analysis of the data. In this process, we recognized that one of our main results (the optimal value of 10-days lag in predictivity) was wrong due to a systematic calculation error, and lead us to a slightly wrong conclusion. We have updated the entire paper, whose staple structure remained nonetheless the same since the rest of the results were not affected by this error. We recognized that 7-days lag is actual the optimal fitting vale for prediction, and the retrieved R2 is lower than originally suggested.

In order to add strength to the proposed evidence, Internal Validation analysis has been performed.

While being sorry for this inconvenient, we remark that the main conclusions have not been deeply affected.

Also, as per Rev. 2 suggestion, we performed a forecast analysis using the retrieved model, showing expected and observed results on a 7-days lag basis.

All data, analysis and models have been updated in supplementary materials as well.

We include here point-by-point responses to the reviewers’ suggestions.

Many thanks and best regards, the authors.

Rev. 2:

Abstract:

  1. The results section in the abstract is very short and provides no actual reporting of raw results.

(1): Abstract results section has been amended as suggested, reporting both R2 and p values as raw results.

Background:

  1. The background section is very general and focuses on describing the Italian context. The authors should focus on existing literature examining relationships between COVID-19 (or other outbreaks) and their relationship to EMS calls.

(1): Thanks for the suggestion. The Introduction has been further expanded and a new section "1.1.3. COVID-19 outbreaks and EMS calls" has been added, as per Rev. 1 suggestion also.

Analyses/Results:

  1. The final analysis model could use clarity. Please confirm that the epidemiological variables assessed for collinearity (see Table 1) resulted in the final selection of using only the ICU admissions variable in the model. I was not sure if the other variables were dropped in the Methods description.

(1): thanks for spotting this. Yes, only ICU admissions' derivative was computed and used in the final model, due to its collinearity with other epidemiological variable and consideration on its data validity robustness expressed in the Discussion section. However, in the methods section it was wrongly implied that all epidemiological variables were nonetheless used in the model. This error has been corrected and the sentence now reads "For Variable (3) (Daily COVID ICU admissions), for each data point y_t, the value of the first-order derivative of its (7)MA, (y_t ) ̇, has been calculated."

  1. Furthermore – perhaps this needs a specific statistical review, but I would assume the authors could use time-series analyses utilizing a regression model versus direct calculation in Excel. There are a lot of equations described in the text although whether this is a valid analysis approach eludes me.

(2): We thank you very much for the hint; had it not been for this, we would not have performed a comprehensive re-analysis of all data, and a crucial mistake would otherwise have passed unseen.

Statistical methods used have been both rigorously described and referenced; moreover, we wish to point that, in supplementary materials, we are providing all data and calculations in clean excel format for any reviewer, reader or researcher to confirm or disprove our results. If this is not deemed sufficient, we are willing to comply with any specific suggestion.

  1. The selection of ICU admission seems to be the appropriate variable used to quantify the epidemic’s trend. This was supported by the collinearity results reported in Table 1.

(3): As answered to point 1, yes, we confirm that ICU admission was the only variable used for trend prediction. Also we believe this is a parameter that makes most sense from a health system management perspective.

  1. This study adds to evidence that EMS calls are predictive of ICU admissions during the COVID-19 pandemic, particularly prior to wide availability of COVID-19 vaccines. Figure 4 is striking in demonstrating how a lag of 10 days between EMS calls and the trend in ICU admissions are analogous. The authors acknowledge how these findings are specific to this epidemic, and these findings may change depending on characteristics of the pathogen (e.g., virulence). However, the authors could comment on how availability of the vaccine may affect the predictive value of EMS calls associated with ICU admissions.

(4): Thanks for the positive feedback. We believe that the most important result is the demonstration that trend calls analysis can be a viable tool in decision-making during an epidemic outbreak, although the specific results of this paper are obviously limited to COVID-19 outbreak. Some of the authors' ideas on vaccination effects have been added in the Discussion section (along with literature references), also as per Rev. 1 indications. Alas, re-analysis showed that 10-days lag was sub-optimal, as the best fitting for the curve occurred between 6 and 11 days; 7-days lag seem to be the one where the two curves are the most near. Both figures, descriptions and plain text have been updated as consequence.

  1. In Figure 4, change in calls and D’ of ICU admissions tracks closely until Spring of 2021. For the months thereafter, what do you make of why EMS calls decreased while ICU admissions were notably increased. The patterns do not seem to track as strongly after this date. Could you explain or talk about this discrepancy?

(5): Thanks for pointing this out. It is true that volume changes are less evident, although it is also true that direction changes keep being similar enough for a positive correlation to be found. Cross-fold validation shows that that period - the 1st semester of 2021 is the one where the two curves diverge the most, to the point of "ruining" the overall model effect. A paragraph with more detailed insight on this has been added in the Results “3.3. Trend of EMS Calls” section, and the whole phenomenon is discussed in the Discussion "4.2 Limitations" section. 

  1. Table 2 show the results of the univariate regression models between change in calls and epidemic trend. Using 10 days as the derivative of ICU admissions looks well supported, but could you break this into 2020 and 2021 along with what you report as combined results for 2020-2021? Would the derivative day still be the same in both years?

(6): Given the change of main results, we chose to perform a separate breakdown analysis in order to test the method. Internal Validation has been performed using  the k-fold cross Validation method. Proper sections have been added in Methods and Results sections.

Discussion:

  1. A strength of this study is that it draws from a large sample of EMS calls in a populous region of Italy. EMS calls are centralized within one management system, ARES. It could be noted that not all calls to EMS result in action taken by EMS, either through responder contact or through transport to a medical facility. What proportion of calls to EMS are transported and did that change throughout the study period? It is likely beyond the scope of this study, but it could be interesting to see if there was a correlation between certain symptoms (e.g., respiratory distress) that were the reason for the EMS call and ICU admissions.

(1) We are moved and flattered by this feedback and by this question, for it shows there is vivid interest in the topic, even beyond the original research scope. Unfortunately, we cannot answer in a comprehensive way to the reviewer on the last part of this suggestion, since data on clinical presentation are not collected for administrative purposes, but for clinical ones, and the Authors do not have consent for Publication: the complexity of situations for consent in emergency scenarios creates a considerable ethical challenge to researchers dealing with such cases. Since the objective of the research were not clinical, no request for additional clinical data was held.

Any claim from the Authors based on patients’ clinical data, and on its strong relation with the epidemic curves, although would provide even more strength to present research, would be illegitimate from a purely scientific standpoint, since it could not be publicly backed by the actual data. For such reasons, we regrettably chose not to make any hint in in this regard, despite likewise believing it would have further improved the paper presentation by adding more strength to the collected evidence; please confront newly inserted subsection "2.5. Privacy, Ethical Committee approval and informed consent" for legislature reference.

This said, we have amended the paper with results on the proportion of EMS calls and ambulance missions, whose data was indeed available to the Researchers. It has been observed that this proportion was quite stable in the observed period, except on the verge of the epidemic waves. The phenomenon has not been object of specific analysis, but it has been nonetheless added in the Results section as secondary finding.

  1. The discussion section is very short (one paragraph) and does not situate the authors findings in the existing literature on this topic. How do the authors findings compare to existing literature utilizing either EMS calls as a predictor of a future epidemic progression – or for other diseases / issues?

(2): Discussion section has been expanded with further comments and some insights on relevant points, also as per Rev. 1 suggestions. Moreover, our results have been compared with the only work we know of that, up to date, is similar enough for a meaningful comparison.

  1. Could the authors examine whether their model and approach either a) extends to other countries or b) whether it can actually predict forward in the pandemic? The pandemic is ongoing, so the authors could make a prediction using their model in this context and then validate their model by comparing it to the actual trend after prediction.

(3) Thanks for the suggestion. We concur that both points are worth being discussed in the manuscript.

  1. a) We believe that the proposed method can be used in any local area with a toll-free EMS call number, since it does not require any prior inference of unknown parameters relative to the population setting. The only thing that should be variable would be the lag value, in relation to the epidemic characteristics of the involved pathogen. Future External Validation studies conducted on different settings, outbreaks and\or populations could add even more evidence on the strength of the proposed method. This has been included in the manuscript’s Conclusions section.
  2. b) As for the point of making actual prediction on current context, and then confront with the actual trend, although forecasting per se was not the main objective of the research, a simple forecasting function has also been proposed. A dedicated section has been added in both Methods and Results section. While the Forecasting results are consistent with the rest of the analysis, we still wish to highlight that our research objective was to prove or disprove the correlation between EMS calls variation and epidemic trend, and its role on strategic decision making, rather to propose yet another prediction model for epidemic diseases.

Minor Points:

  1. Check for spelling errors throughout the manuscript. For example, “spleen” interpolation (line 112) should probably be spline interpolation. Line 74 “Quasi-sperimental” is likely Quasi-experimental?

(1): Several minor spelling errors have been corrected, among those the two identified above.

  1. Confirm population estimate of Lazio region. It says over 5.5 million on line 40 and over 6 million on line 78-79, and almost 6 million on line 281.

(2): Line 40 estimation has been corrected and put in line with current National Statistical Institute data; updated reference.

  1. HES was not defined. I assume this is Health Emergency Services on line 69. Consider calling them emergency calls or calls. “EMS calls” is also used, see line 123. There are many abbreviations and terms using in reference to calls in this paper.

(3): "HES" was indeed a wrong acronym. Corrected to EMS in several instances.

Round 2

Reviewer 2 Report

The authors have answered all of my questions. 

This manuscript is a resubmission of an earlier submission. The following is a list of the peer review reports and author responses from that submission.